# Differential interaction forces govern bacterial sorting in early biofilms

Enno R Oldewurtel, Nadzeya Kouzel, Lena Dewenter, Katja Henseler, Berenike Maier*

Department of Physics, University of Cologne, Cologne, Germany

**Abstract** Bacterial biofilms can generate micro-heterogeneity in terms of surface structures. However, little is known about the associated changes in the physics of cell–cell interaction and its impact on the architecture of biofilms. In this study, we used the type IV pilus of *Neisseria gonorrhoeae* to test whether variation of surface structures induces cell-sorting. We show that the rupture forces between pili are fine-tuned by post-translational modification. Bacterial sorting was dependent on pilus post-translational modification and pilus density. Active force generation was necessary for defined morphologies of mixed microcolonies. The observed morphotypes were in remarkable agreement with the differential strength of adhesion hypothesis proposing that a tug-of-war among surface structures of different cells governs cell sorting. We conclude that in early biofilms the density and rupture force of bacterial surface structures can trigger cell sorting based on similar physical principles as in developing embryos.

## Introduction

Physical interactions and in particular mechanical forces are involved in sorting of different cell types. These requirements have been established in the field of embryonic development (*Gonzalez-Rodriguez et al., 2012*). Cells must divide, change shape, adhere to each other, migrate, disperse, or cluster. To complete these processes, they modulate their physical properties including rigidity, position, motility, and strength of adhesion (*Fagotto, 2014*). The development of bacterial biofilms resembles eukaryotic development in various aspects including genetic programs that are controlled by master regulators (*Monds and O'Toole, 2009*). However, it is unclear to which extent physical forces are involved in determining the architecture of biofilms.

An early approach for understanding cell sorting on the basis of physical interactions was the differential adhesion hypothesis (*Steinberg, 1963*). This hypothesis states that a population of cells will sort into spatially separated subpopulations, if the strength of cell–cell adhesion differs between cell types. The biophysical explanation for this sorting is the tendency of cell clusters to minimize their surface, similar to the minimization of surface tension in liquids. Depending on the strength of cohesion between cells of the same kind and the strength of adhesion between different kinds of cells, the model predicts perfect mixing, encasement, partial segregation, and full segregation of cells (*Steinberg, 1963*; *Graner and Glazier, 1992*). Some of these morphotypes have been observed experimentally (*Steinberg and Takeichi, 1994*; *Foty and Steinberg, 2004*). Recent experiments indicate that equilibrium descriptions are not sufficient for describing the process of cell sorting during development; they provide evidence that active force generation by cells is involved (*Landsberg et al., 2009*; *Gonzalez-Rodriguez et al., 2012*; *Kashef and Franz, 2015*; *Mao and Baum, 2015*). In particular, the differential interfacial tension hypothesis introduces cortical contractility in addition to adhesion (*Brodland and Chen, 2000*). Interestingly, Harris proposed another hypothesis, the differential strength of adhesion hypothesis (DSAH) (*Harris, 1976*). He envisions a tug-of-war of cells actively moving along each other by retractions of cellular extensions pulling the cell body into

*For correspondence: berenike.maier@uni-koeln.de

Competing interests: The authors declare that no competing interests exist.

**eLife digest** Communities of bacterial cells can live together embedded within a slime-like molecular matrix as a biofilm. This allows the bacteria to hide from external stresses. A single bacterium can replicate itself and develop into a biofilm, and over time the bacterial cells in specific regions of the biofilm will start to interact with their neighbors in different ways. These interactions occur via structures on the surface of the bacterial cells, and the differences in these interactions resemble those that occur as cells specialize during the development of animal embryos. Previous research into embryonic development has shown how differences in the physical interactions between embryonic cells are essential for sorting the cells into their correct locations and shaping the embryo. However, little is known about which processes govern the development of biofilms.

Now, Oldewurtel et al. have asked whether differences in the physical interactions between bacteria trigger cell sorting during the early stages of biofilm development. The experiments involved measuring the force required to break the cell–cell connections (called the 'rupture force') in biofilms of a bacterium called *Neisseria gonorrhoeae*. Oldewurtel et al. found that, in agreement with previous predictions, physical interactions were important for sorting bacterial cells into clusters based on the structures on their surfaces. Bacterial cells actively pull on the surface structures of their neighbors, which allows the cells to sort themselves in a tug-of-war fashion. This means that a cell will move in the direction where it can pull the strongest (i.e., in the direction where the rupture force is highest).

While bacteria and embryos use different molecules to generate these pulling forces, these findings indicate that the basic physical principles are similar in both systems. One of the next challenges will be to evaluate how biofilms might benefit from the structures that develop due to cell sorting.

the direction of the strongest and least breakable bonds. Cells with lower bond strength would be squeezed outward to the peripheral layer.

Sorting of cells also occurs during bacterial biofilm development. Cell–cell interactions and biofilm architecture are often governed by cell appendages with polymeric organization. One of the most ubiquitous appendages is the type IV pilus (T4P) (*Berry and Pelicic, 2015*). It mediates surface attachment, surface motility, and its retraction can generate high mechanical force (*Maier and Wong, 2015*; *Maier, 2013*). The T4P is a helical polymer consisting mainly of the major subunit PilE (*Craig et al., 2004*). The major subunit can be post-translationally modified by O-linked glycosylation or by phosphoform-modifications. Cell sorting based on T4P has been observed in different experimental setups. *Vibrio cholerae* can express a T4P-like system and generate toxin co-regulated pili. Cells lacking the major pilin of the toxin co-regulated pilus did not integrate into wt microcolonies (*Kirn et al., 2000*). Mutations in the major pilin subunits induced different pilus morphology and affected the size of pilus bundles. However, cell sorting based on these mutations was not reported (*Kirn et al., 2000*). Differential fluorescence labeling of *Pseudomonas aeruginosa* showed that cells with T4P-dependent motility form a cap on top of non-motile stalks formed by cells that lacked the gene for the major pilin subunit (*Klausen et al., 2003*). Swarming of *P. aeruginosa* is generated by a single flagellum, but moderated by polar T4P (*Anyan et al., 2014*). When T4P production was suppressed, swarming motility increased considerably, most likely due to reduced cellular clustering mediated by pili. When piliated and non-piliated cells were co-cultured, non-piliated cells dominated the swarm edge. *Neisseria meningitidis* upregulates expression of a phosphotransferase that is required for phospho-form modification of the major pilin upon attachment to endothelial host cells (*Chamot-Rooke et al., 2011*). Simulations of pilus bundling suggest post-translational modifications affecting the physical interactions between T4P (*Chamot-Rooke et al., 2011*). All of these studies imply that T4P can generate a rich diversity of sorting phenomena. However, in contrast to embryonic development, the biophysical basis underlying cell sorting in bacterial biofilms remains elusive.

Here, we test the hypothesis that the physical interactions between bacteria govern the morphology of mixed bacterial cell clusters. The T4P of *Neisseria gonorrhoeae* (gonococcus) was used as a model system for systematic variation of cell–cell interactions. We genetically engineered strains with different densities of pili and their abilities to generate force. Moreover, the breakage-force between pili of

different cells could be fine-tuned by replacing a gene responsible for pilin post-translational modification. We found that on agar plates, where the dynamics are mostly determined by steric interactions caused by cell division, cells with the lowest pilus density and cells with the lowest pilus-breakage force sorted towards the colony boundary. Sorting was prominent at the front of expanding colonies but within the bulk of the colony sorting was incomplete. In liquid environment where cells actively move by pilus retractions, sorting correlating with pilus density and pilus-breakage force was nearly complete and the resulting morphotypes followed the DSAH. Active pilus retraction was essential for cell sorting, suggesting that the cells sort by a tug-of-war between cells.

## Results

### The T4P as a tool for controlling bacterial cell–cell interactions

*N. gonorrhoeae* and their type IV pili served as a model system for addressing the hypothesis that mechanical interactions control sorting and morphology of mixed clusters (aka microcolonies). Gonococci are peritrichously piliated, that is, they generate pili at random locations (*Holz et al., 2010*). Because T4P are necessary for microcolony formation, variation in the breakage force between pili directly affects cell–cell interactions. There is preliminary evidence that gonococcal microcolonies show surface tension-like behavior. Microcolonies formed by gonococci with retractile pili are spherical (*Higashi et al., 2009*). Upon fusion, two microcolonies rapidly form a sphere with larger radius as expected for liquid drops (*Dewenter et al., 2015*). Importantly, the retraction of T4P generates mechanical force (*Maier et al., 2002*). Multiple pili can coordinate through a tug-of-war mechanism leading to surface motility (*Marathe et al., 2014*). Hence, cell movement inside clusters is conceivable. These properties could support cell sorting as proposed by the DSAH (*Harris, 1976*).

To test the hypothesis of segregation of cells depending on receptor–ligand pair densities, we engineered gonococcal strains with different pilus densities. To start with, we generated a gonococcal strain that had the gene for the major pilin subunit, *pilE*, replaced by *gfpmut3* and *kan* (*pilE::gfpmut3 kan*), henceforth called *P− green* strain (*Table 1*). This strain did not form clusters nor did it interact with piliated bacteria. Thus, within our detection limit, the *P− green* bacteria show no interactions. Next, we used a strain with three copies of the *pilE* gene, *P++* (*igA1::pilE pilE ermC*). This strain contains two copies of the *pilE* gene in addition to the native copy, increasing the pilus density compared to the wild type (*Holz et al., 2010*). Since the pilins are unaltered, the rupture forces between individual pili are expected to be equal to the wt forces. However, since the density of pili is higher, the total attractive force between two bacteria is higher.

**Table 1**. Strains used in this study

| Strain | Relevant genotype | Source/Reference |
|---|---|---|
| VD300 | *wild type, opa- selected* | – |
| Ng105 *P+ green* | *igA1::P$_{pilE}$ gfpmut3 ermC* | This study |
| Ng106 *P+ red* | *lctP: P$_{pilE}$ mcherry aadA:aspC* | This study |
| Ng081 *P− green* | *pilE::P$_{pilE}$ gfpmut3 kan* | This study |
| Ng095 *G− green* | *pglF::PpilE gfpmut3 kan* | This study |
| Ng109 *P++* | *igA1::pilE pilE ermC* | This study |
| Ng110 *P++ red* | *igA1::pilE pilE ermC; lctP: P$_{pilE}$ mcherry aadA:aspC* | This study |
| Ng118 *Q− green* | *pilQ::m-Tn3cm; igA1::P$_{pilE}$ gfpmut3 ermC; recA6ind(tetM);* | This study |
| Ng116 *P+ red** | *lctP: P$_{pilE}$ mcherry aadA:aspC; recA6ind (tetM);* | This study |
| Ng119 *T− green* | *pilT::m-Tn3cm; igA1::P$_{pilE}$ gfpmut3 ermC* | This study |
| Ng120 *T− red* | *pilT::m-Tn3cm; lctP: P$_{pilE}$ mcherry aadA: aspC* | This study |
| Ng121 *T−G− green* | *pilT::m-Tn3cm; pglF::PpilE gfpmut3 kan* | This study |

In the next step, we investigated the effect of pilin post-translational modification on the breakage force between pili. Wt gonococci bear a disaccharide composed of a hexose residue linked to a proximal 2,4-diacetamido-2,4,6-trideoxyhexose (HexDATDH) at serine 63 (*Hegge et al., 2004*). PglF is involved in membrane translocation of lipid-attached carbohydrates (*Hegge et al., 2004*). Deletion of *pglF* strongly reduces O-linked pilin glycosylation. Hence, we generated strain *G– green* by replacing *pglF* with *gfpmut3* (*Table 1*).

When interfering with the post-translational modification of the pilins, we expect that the rupture force between pili is affected. We used laser tweezers for measuring the rupture forces, employing a previously established assay (*Dewenter et al., 2015*). The surface was coated with a crude pilus preparation extracted from the strain of interest. Subsequently, single bacteria (monococci) were caught in a laser trap and held close to the T4P-coated surface (*Figure 1A*). The forces applied by the optical trap kept the cell body at the center of the trap. Type IV pili can generate very large force by retraction when they bind to surfaces (*Maier et al., 2002*). Binding of T4P to the pilus-coated surface and subsequent retraction led to a deflection of the cell body. Upon breakage of the bond between the retracting pilus and the pili on the surface, the cell body is pulled back into the center of the laser trap. Thus, the maximum force between pili attached to the bacterium and the pili attached to the surface can be measured. Henceforth, we will call this force rupture force. The distribution of rupture forces of wt bacteria pulling on wt pili was in agreement with a Gaussian distribution with a maximum at $F_{wt/wt} = (39 \pm 1)$ pN (*Figure 1B*). The distribution of rupture forces generated by *G– green* was shifted to higher values of $F_{G-/G-} = (46 \pm 1)$ pN. Most interestingly, the force generated by *G– green* onto the surface coated with wt pili, $F_{G-/wt} = (25 \pm 1)$ pN, was considerably lower than the interaction forces between pili of the same kind. Moreover, we monitored the frequency at which bacteria pulled onto the pilus-coated surface and found no significant difference between the strains (*Figure 1C*), confirming that pilus–pilus rupture force and not the frequency of pilus retractions was responsible for

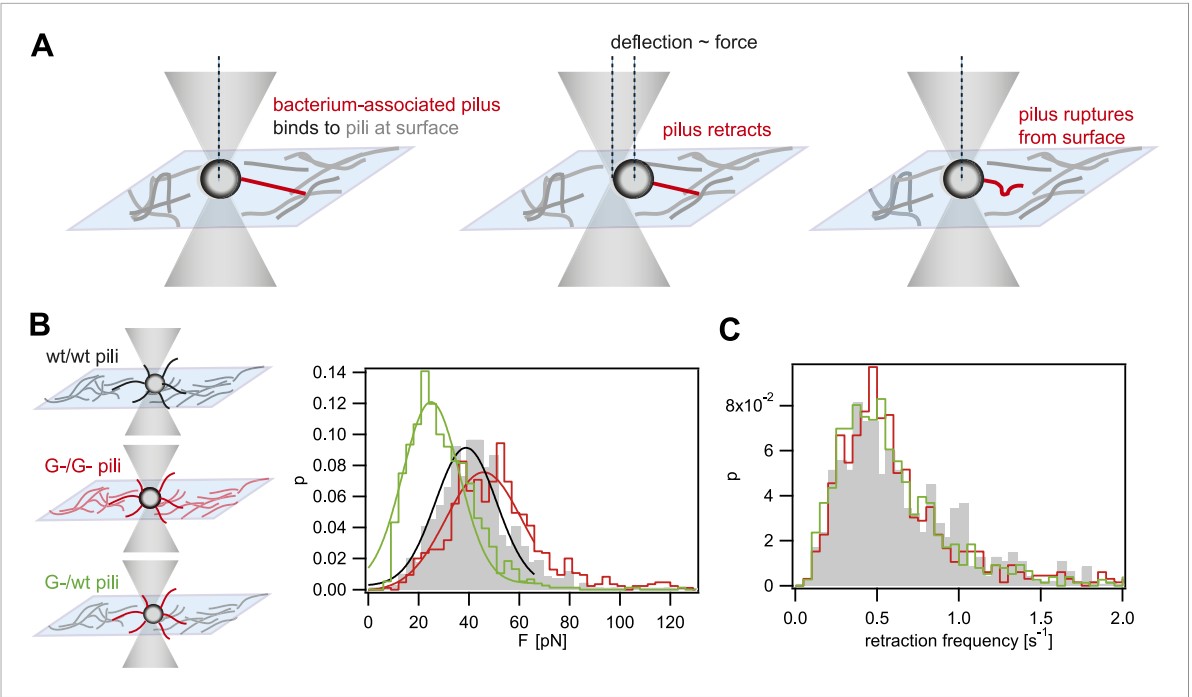

**Figure 1.** Rupture forces between T4P. (**A**) Principle of force measurement. The surface was coated with pili. A single monococcus was trapped in an optical trap. One or multiple pili can bind to pili at the surface. When they retract they deflect the bacterium from the center of the laser trap. The deflection is proportional to the force acting on the bond between the bacterium-associated pili and the pili at the surface. When the bond between the pili is ruptured, the bacterium moved back to the center of the trap. For each retraction event, the maximum force generated prior to rupture was registered as the rupture force. (**B**) Distribution of rupture forces. Full lines: Gaussian fit for $F < 65$ pN. For $F > 65$ pN, the linear regime is exceeded and forces are overestimated. (**C**) Distribution of retraction frequencies. Gray bars: wt bacteria on surface coated with wt pili. Red line: *G– green* bacteria on surface coated with *G– green* pili. Green line: *G– green* bacteria on surface coated with wt pili.

the altered cell–cell interaction in response to the loss of pilin glycosylation. Taken together, we found that $F_{G-/G-} > F_{wt/wt} > F_{G-/wt}$.

In summary, the breakage force between gonococci is fine-tuned by post-translational modification of pilin. Together with mutants of varying T4P density, our T4P toolbox will enable us to test how mechanical forces between cells affect cell sorting.

## Bacteria with lowest density of pili segregate to the front of expanding microcolonies

We developed a tool for directly visualizing the spatio-temporal dynamics of segregation between different strains. Gonococci were inoculated onto agar plates at a density low enough to ensure microcolonies arising from individual bacteria. Time-lapse microscopy was used for following the dynamics of their off-spring. This way, we ensured that the population was clonal at the start of the experiment. New clones inside these populations with modifications in T4P were generated as follows. *N. gonorrhoeae* are naturally competent for transformation, which is the import and inheritable integration of DNA from the environment (*Burton and Dubnau, 2010*). Transformation allows for fluorescent labeling in conjunction with manipulation of genes affecting T4P. A control strain was generated which became fluorescent without affecting piliation by integrating the fluorescence reporter into a non-essential gene (*igA1::gfpmut3 ermC*), henceforth called *P+ green* strain (*Figure 2A*). Genomic DNA (gDNA) from this strain was isolated and applied onto an agar plate. Each cell of the expanding wt colony can integrate *gfpmut3 ermC* into the *igA1* locus and become fluorescent with a certain probability. Since the fluorescent marker was inheritably integrated, all progeny of a single transformant was fluorescent (*Video 1*). In *Figure 2B*, the first *P+ green* cell was detected at 5 hr. As expected for range expansion experiments (*Hallatschek et al., 2007*), little mixing between *P+ green* and wt was observed. Instead, *P+ green* formed a sector. This observation suggests that the mobility of the bacteria in this assay was mostly determined by steric interactions caused by cell division. We note that T4P-mediated surface motility was observed only within 1–3 bacterial diameters at the expanding front (*Video 2*).

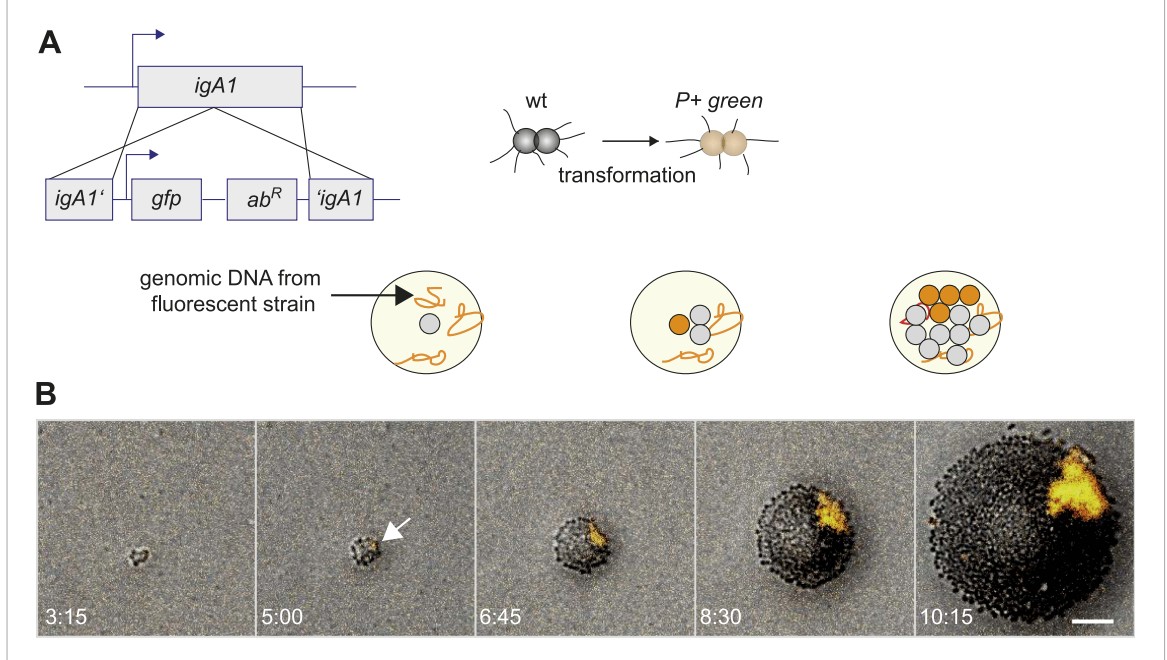

**Figure 2**. Assay for direct visualization of spatio-temporal dynamics of a new clone and its progeny within an expanding microcolony. (**A**) Strategy for visualization of a new clone. Genomic DNA (gDNA) from strain *P+ green* in which *gfpmut3* recombines into *igA1* is spread on the agar surface. When a single bacterium imports and integrates the DNA, then the bacterium becomes fluorescent. Type IV pilus (T4P) is unaffected. (**B**) Time-lapse of de novo occurring *P+ green* clone (orange) within an expanding wt colony. Scale bar: 10 μm. The arrow denotes the time point when fluorescence is detectable.

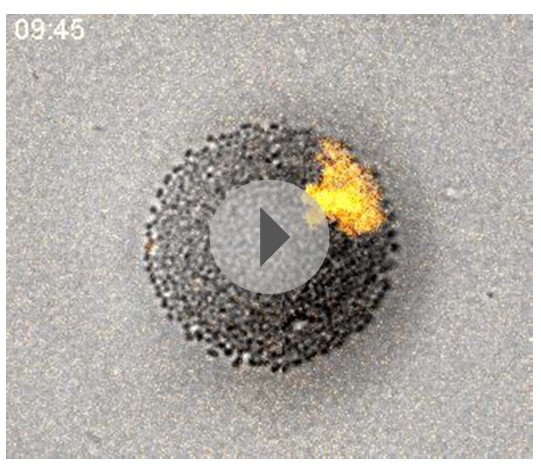

**Video 1.** Spatio-temporal dynamics of de novo occurrence of a P+ clone and its offspring. Chromosomal DNA from strain *P+ green* is spread on the agar surface. Upon import and integration of DNA into a single bacterium, pili are lost and the bacterium becomes fluorescent. Time-lapse (Δt = 15 min) of de novo occurring *P+ green* clone within an expanding colony. Brightfield image and fluorescence images were merged.

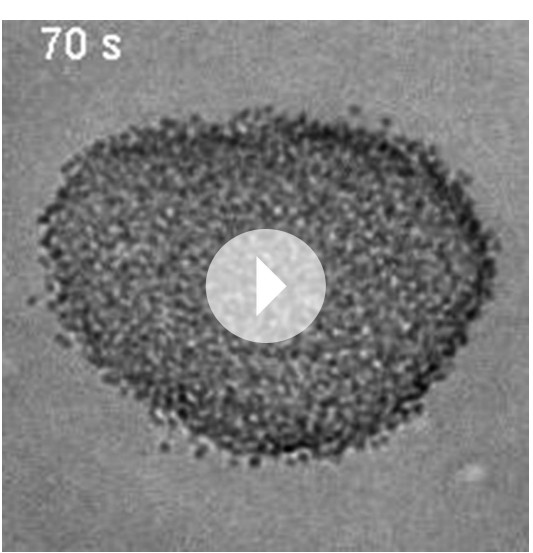

**Video 2.** Mobility of bacteria on agar plate. *P+ green* bacteria were inoculated onto an agar plate. Images were acquired at Δt = 10 s. Bacteria residing within the bulk of the microcolony were immobile at this time scale. At the front, individual bacteria are motile. This motility depends on active T4P retraction and was not observed for non-piliated bacteria. Scale bar: 10 μm.

Next, we investigated whether loss of T4P caused segregation. gDNA from the *P− green* strain was isolated and applied onto an agar plate. The replacement of the pilin gene by *gfpmut3* resulted in loss of pili and gain of fluorescence (*Figure 3A*). P− bacteria arising close to the expanding front were expelled from the bulk of the colony and their offspring spread rapidly along the front, encircling the entire colony (*Figure 3C,D*, *Video 3*).

Furthermore, we assessed whether reduction of pilus density (i.e., the average number of pili per cell) leads to a segregation of variants. DNA from *P+ green* cells was isolated and spread onto an agar plate (*Figure 3B*). Hyperpiliated *P++* gonococci were seeded onto the plate. The integration of DNA from *P+ green* into this strain replaces the two additional *pilE* copies by a *gfpmut3* gene, and as a consequence, the pilus density reduces to wt level. We found that the *P+ green* cells residing close to the expanding front were expelled from the colony and encased the *P++* colony (*Figure 3E*).

So far, the experiments showed that cell sorting occurred at the front of the expanding microcolonies but sorting was incomplete when considering the entire microcolony. We analyzed how the probability that *P− green* gonococci moved towards the front was dependent on their location within the colony. For systematic analysis of the dynamics of a de novo occurring *P− green* clone and its offspring, we automatically detected the contour of a sector of fluorescent bacteria generated by a single variant (*Figure 5—figure supplement 1*). The location of the fluorescent pixel closest to the expanding front was determined as a function of time (*Figure 4A*). *P− green* bacteria residing directly at the front remained at the front and tended to form blebs (*Figure 3C*). *P− green* bacteria closer than 3 μm to the front showed a probability of more than 50% to move towards the front (*Figure 4C*). At times, hopping of gonococci to the front was observed (e.g., red line in *Figure 4A*). This probability decreased as a function of distance. In contrast, the control strain *P+ green* only showed a probability of 50% of residing and staying at the front (*Figure 4B,C*), which decreased as a function of distance and stayed significantly below the *P− green* probability for reaching the front.

Our data show a strongly increased probability of reaching the front for non-piliated bacteria. T4P are responsible for cell–cell interaction in gonococcal colonies. Thus, the mobility of *P−* gonococci within a P+ colony might be increased. To this end, we measured the spatial variance $\sigma^2$ of all offspring of a single *P− green*

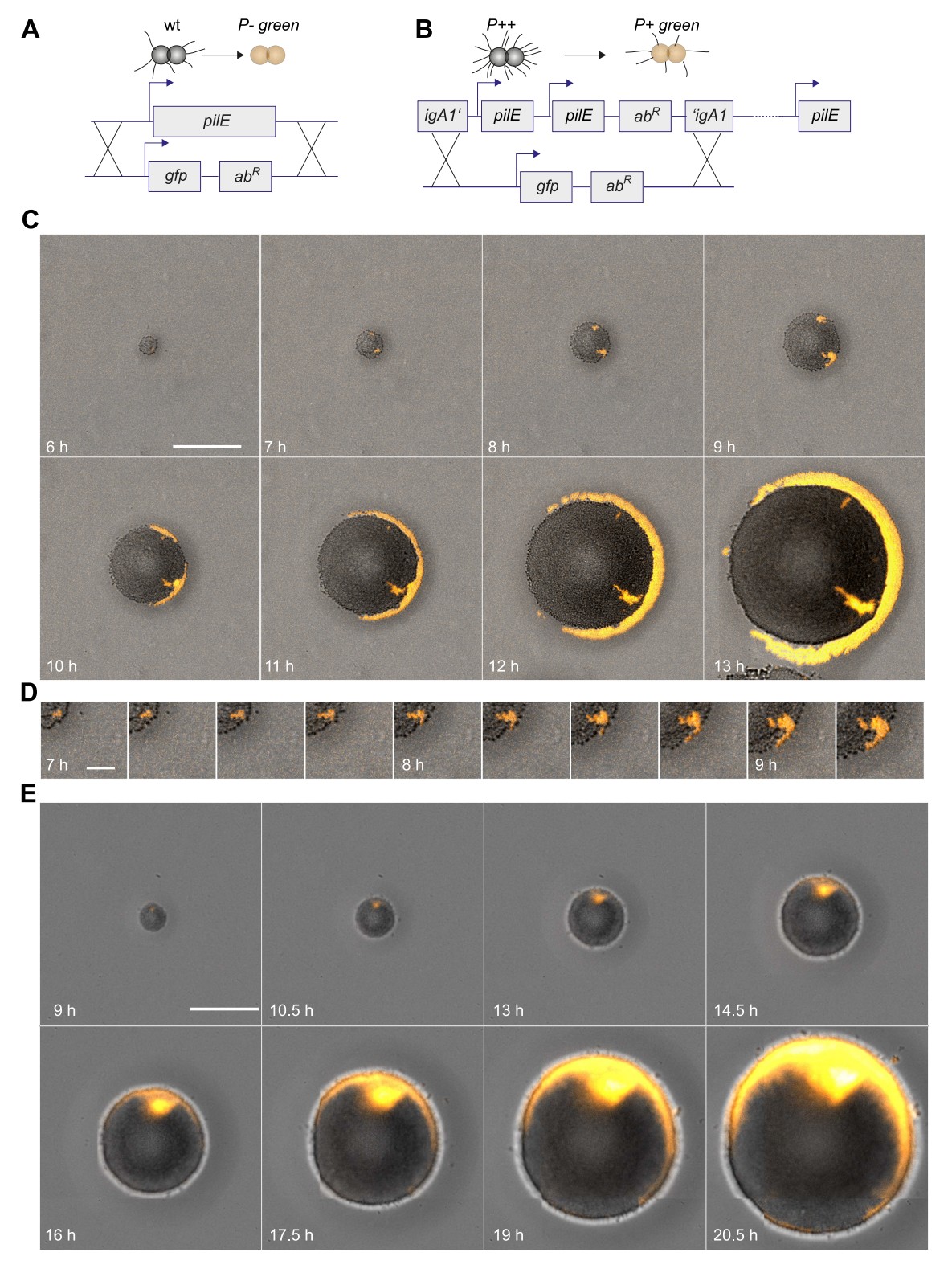

**Figure 3**. Spatio-temporal dynamics of de novo occurrence of clones with reduced pilus density and their offspring. (**A**) Strategy for visualization of pilus-loss in a single bacterium. gDNA from strain *P− green* in which *pilE* is replaced by *gfpmut3* is spread on the agar surface. Upon import and integration of DNA into a single bacterium, pili are lost and the bacterium becomes fluorescent. (**B**) Strategy for visualization of reduction of pilus-density. gDNA from *P+ green* strain is spread on the agar surface. Integration of DNA into a single hyperpiliated *P++* bacterium carrying two additional copies of *pilE* in the

*Figure 3 continued on next page*

*Figure 3 Continued*

*igA1* locus led to the pilus density decreasing to wt level and to the acquisition of fluorescence. (**C**) Time-lapse of *de novo* occurring *P− green* clone (orange) within an expanding colony. Scale bar: 50 µm. (**D**) Detail of (**C**). Scale bar: 10 µm. (**E**) Time-lapse of *de novo* occurring *P+ green* clone (orange) within an expanding colony of *P++*. Scale bar: 50 µm.

transformant compared to the variance of the offspring of a single *P+ green* transformant (*Figure 5*) inside growing microcolonies. The variance is expected to increase because bacteria are both mobile, but also increase in number within the colony. By rescaling to the size of the microcolony, we found that the variance still increased (*Figure 5—figure supplement 1C,D*), suggesting that bacteria are mobile.

Next, we tested whether the mobility of *P− green* within a wt microcolony was increased. For this analysis, only variants that are not touching the front are analyzed (*Figure 5A–C*). Colony 2 (green) shows an example of a sector touching the front at t = 7.5 hr. Later time points of such sectors were automatically removed from the analysis. As expected, the variance increased as a function of time (*Figure 5B*). Different growth rates might obscure the effect of differential physical interactions on cell. Hence, the generation time of gonococci expressing different amounts of the major pilin *pilE*, which results in varying densities of pili per cell (*Long et al., 2001*; *Holz et al., 2010*) (*Table 2*, 'Materials and methods'), was measured. The level of pilin expression strongly affected the growth rate. The *P+ green* strain had a generation time of $t_g = 49 \pm 3$ min. The non-piliated *P− green* strain had a significantly lower generation time ($t_g = 41 \pm 1$ min), whereas the growth rate of the hyperpiliated *P++ red* strain showed an increased generation time ($t_g = 56 \pm 4$ min). In our mobility analysis, we compensated for different generation times, by comparing the variance of *P− green* with *P+ green* as a function of the number of offspring (*Figure 5C,D*). We found that the variance of the *P− green* cells increases more strongly as a function of the number of offspring than the variance of *P+ green* cells, indicating an increased mobility of *P− green* within a colony of wt bacteria.

Furthermore, segregation of strains with different pilus densities but similar growth rates was assessed. We generated the gonococcal strain *P^Q− green* with a deletion of *pilQ*. Since the PilQ proteins form the pore through which the pilus is exported, this results in a *P−* phenotype. The generation time of *P^Q− green* was slightly higher than the piliated strain *P+ red\**. We inoculated non-piliated *P^Q− green* bacteria at a low density together with a higher density of piliated *P+ red\** bacteria (*Figure 5—figure supplement 2A,B*). Similar to *P− green*, the *P^Q− green* close to the front of the expanding colony was expelled from *P+ red\** (*Figure 5—figure supplement 2A*). Once reaching the front, *P^Q− green* expanded along the front and encircled the bulk of *P+ red\** cells. This control experiment confirms that loss of pili induces segregation independently of decreased generation time.

In summary, bacteria with the lowest pilus density segregate to the front of mixed expanding microcolonies. The sorting efficiency strongly depends on the location of the less piliated clone, with highly efficient sorting close to the front.

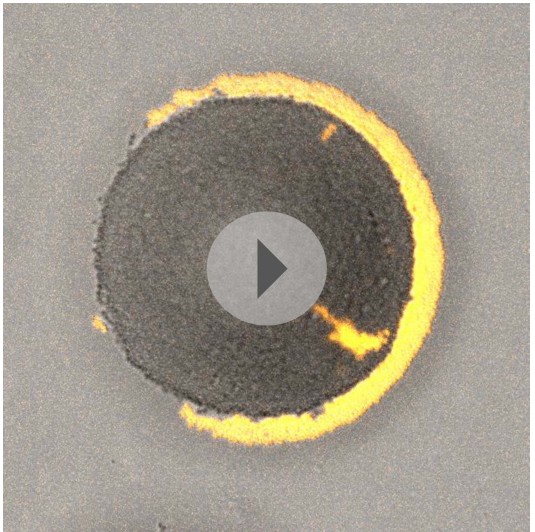

**Video 3.** Spatio-temporal dynamics of *de novo* occurrence of a P− clone and its offspring. Chromosomal DNA from strain *P− green* in which *pilE* is replaced by *gfpmut3* is spread on the agar surface. Upon import and integration of DNA into a single bacterium, pili are lost and the bacterium becomes fluorescent. Time-lapse (Δt = 15 min) of *de novo* occurring *P− green* clone within an expanding colony. Brightfield image and fluorescence images were merged.

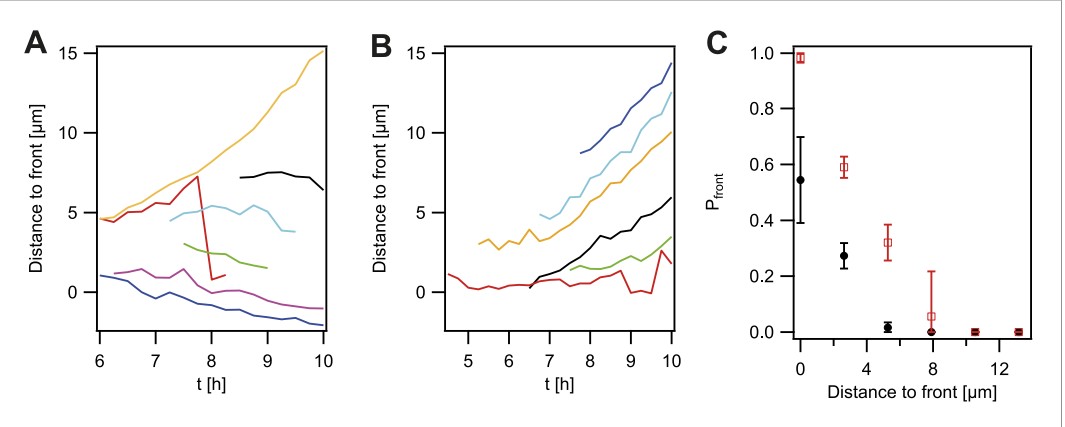

**Figure 4**. Probability that the offspring of a newly arisen P− gonococcus is moving towards the expanding front. (**A**) Location of the bacterium closest to the front of the *P− green* subpopulation as a function of time. (7 out of 148 traces are shown.) (**B**) Location of the bacterium closest to the front of the *P+ green* subpopulation as a function of time. (6 out of 307 traces are shown.) (**C**) Probability that the offspring moves towards the front as a function of its distance from the front for *P− green* (red) and *P+ green* (black). Error bars: standard deviation of three independent experiments weighted with the number of traces per experiment.

## Bacteria with post-translationally modified pili dominate the front of expanding colonies

Pilin glycosylation reduces the rupture force between pili (*Figure 1*). In the context of the DSAH, we predict that bacteria with glycosylated pili segregate to the front of an expanding microcolony. In strain *G− green*, the gene encoding for the flippase, *pglF* (*Aas et al., 2007*), was replaced by a gene encoding for a green fluorescent protein (*pglF::gfpmut3 kan*). Deletion of *pglF* reduces pilin glycosylation severely (*Aas et al., 2007*). Importantly, the generation time of $t_g = 50 \pm 2$ min of this strain was comparable to the wild-type strain with reporter (*Table 2*). If cell sorting was observed, then we could attribute it to altered pilus–pilus interactions.

gDNA of *G− green* was spread onto an agar plate and individual wt cells were inoculated. Wt gonococci that integrated the fluorescence reporter simultaneously lost the glycosylation of their pili. The sectors formed by *G− green* bacteria could not be discerned from sectors formed by *P+ green*. Therefore, we inverted the scenario by inoculating *G− green* gonococci together with gDNA from wt bacteria onto an agar plate (*Figure 6A*). *G− green* gonococci that incorporated the extracellular DNA simultaneously lost their fluorescence and gained the ability to glycosylate their pili. Here, glycosylated (wt) bacteria segregated at the front of the expanding colony (*Figure 6B*).

We conclude that gain of O-linked pilin glycosylation causes segregation of gonococci. The bacteria with glycosylated pili dominate the front of the expanding colony. The underlying cause of sorting is most likely the difference in pilus–pilus rupture forces.

## Variation of pilus density and pilin post-translational modification cause cell sorting in liquid

The agar plate assay described above is highly useful for visualizing the spatio-temporal dynamics of mixed microcolonies. However, cell sorting is restricted to the expanding front, most likely because the cell mobility is low within the bulk of the colony. We used a complementary assay for characterizing the degree of cell sorting. In liquid environment, T4P retractions ensure rapid cell movement and microcolonies assemble, disassemble, and change their shape within minutes (*Dewenter et al., 2015*). Therefore, cell sorting based on different physical interactions can be investigated neglecting any potential segregation resulting from different growth rates.

We mixed bacteria with different T4P and fluorescent markers in liquid and used confocal microscopy for imaging. As control, a suspension of *P+ red* and *P+ green* was used. They formed spherical and well-mixed microcolonies (*Figure 7A*, *Video 4*).

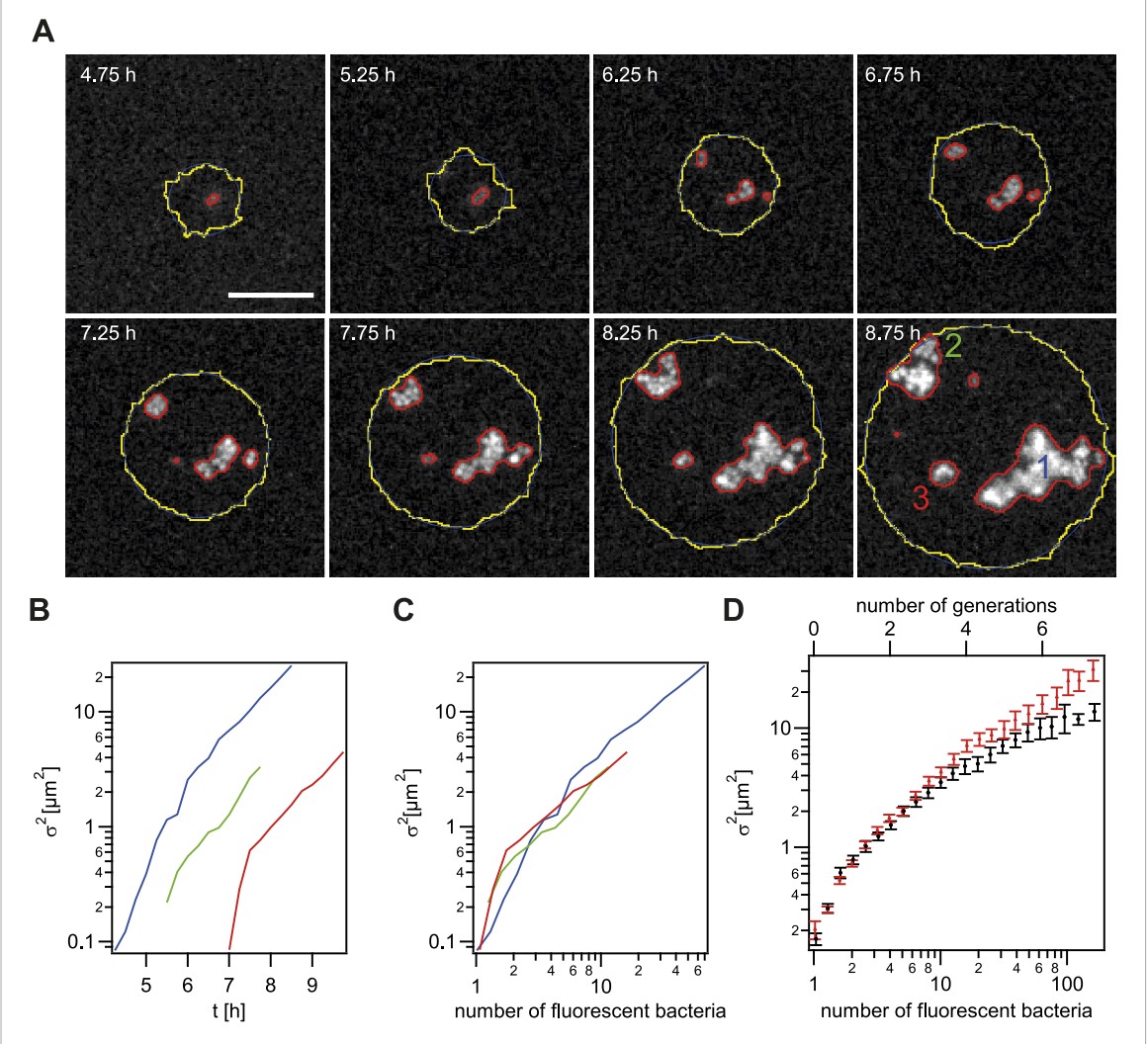

**Figure 5**. Spreading of new clones within an expanding colony. gDNA from *P− green* was spread on the agar plate and wt cells were seeded. (**A**) Fluorescence time lapse. Yellow lines: front of the expanding colony. Red outlines: boundaries of sectors formed by the offspring of a single transformant. Scale bar: 10 µm. Spatial variance $\sigma^2 = \sigma_x^2 + \sigma_y^2$ of three sectors as a function of (**B**) time and (**C**) number of offspring (fluorescent bacteria per sector). The colors correspond to the colors of the numbers at 8.75 hr. (**D**) Average variance of *P− green* (red) and *P+ green* (black) within the colony as a function of the number of offspring $N$. Error bars: standard error as obtained from >50 sectors for each condition.

The following figure supplements are available for figure 5:

**Figure supplement 1**. Analysis of the dynamics of the offspring of a single transformant.

**Figure supplement 2**. Segregation dynamics of non-piliated $P^Q−$ *green* from $P^+$ *red\**.

When *P+ red* and *P− green* were mixed, the wt bacteria formed spherical microcolonies (*Figure 7B*, *Video 5*), while the *P− green* bacteria did not show interaction amongst themselves or with wt bacteria.

Both the DAH and the DSAH predict that if cells with different densities of the same adhesins are mixed, then the cells with higher density will be surrounded by cells with lower density. To test this hypothesis in our bacterial model system, *P++ red* and *P+ green* were mixed. We found that *P+ green* formed a shell surrounding *P++ red* (*Figure 7C*, *Video 6*). This behavior is consistent with stronger physical interactions among *P++* bacteria due to their high-pilus density, lower interactions among *P+* bacteria due to lower pilus density, and intermediate interactions at the interface between *P++ red* and *P+ green*.

**Table 2**. Generation times

| Strain | Generation time | Total number of evaluated colonies |
|---|---|---|
| ig:A1:P$_{pilE}$ gfpmut3 ermC (P+ green) | 48.6§ ± 2.7† min | 76 |
| igA1::pilE pilE ermC; lctP: P$_{pilE}$ mcherry aadA:aspC (P++ red) | 55.6 ± 3.7 min | 33 |
| pilE::gfpmut3 kan (P− green) | 41.1 ± 0.8 min | 84 |
| pglF::P$_{pilE}$ gfpmut3 kan (G− green) | 49.7 ± 2.0 min | 14 |
| pilQ::m-Tn3cm; igA1::P$_{pilE}$ gfpmut3 ermC; recA6ind(tetM) (P$^Q$− green) | 51.1 ± 0.3 min | 280 |
| lctP: P$_{pilE}$ mcherry aadA:aspC; recA6ind (tetM) (P+ red*) | 49.4 ± 0.6 min | 211 |

§Average value of results obtained from at least three independent experiments. Each experiment comprises the analysis of several individual colonies.

†Corrected sample standard deviation from at least three independent experiments.

The *pglF*-deletion strain *G− green* showed differential rupture forces, namely $F_{G-/G-} > F_{wt/wt} > F_{wt/G-}$ (**Figure 1**). In a mix of *P+ red* and *G− green*, *P+ red* formed crescents surrounding *G− green* (**Figure 7D**, **Video 7**). This morphotype agrees remarkably well with the prediction of the DAH (**Steinberg, 1963**).

We conclude that bacteria sort dependent on pilus density and pilin post-translational modification. The morphologies of the mixed colonies can be inferred from the rupture forces between the pili.

## Active force generation by T4P is required for the defined morphology of mixed clusters

All experimental results show that bacterial cell sorting can be explained on the basis of pilus–pilus rupture forces. Cell sorting on agar plates where motility is low hints on the importance of cell mobility for sorting. The observation of almost complete sorting in liquid environment is in agreement with both the passive DAH and the DSAH, the latter includes cellular force generation. We addressed the role of force generation in cell sorting, by transferring T4P-related mutations into a *pilT−* background (*T−*). *pilT* encodes for the T4P retraction ATPase PilT and is essential for pilus retraction (**Wolfgang et al., 1998**). In *T−* cells, T4P can form, but cannot retract, nor generate force. As a consequence, bacteria are unable to perform active movement.

*T− red* and *T− green* were mixed and imaged with confocal microscopy (**Figure 8A**). Unlike wt gonococci, the clusters did not round up to form spherical microcolonies. Furthermore, red and green

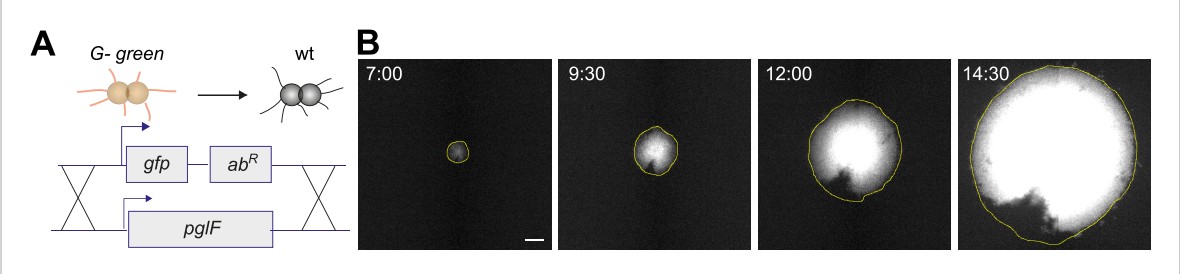

**Figure 6**. Spatio-temporal dynamics of de novo occurrence of clones with the ability to glycosylate pili. (**A**) Strategy for visualization of gain of glycosylation in a single bacterium. gDNA from wt is spread on the agar surface. Upon import and integration of wt DNA into a single *G− green* bacterium, both the fluorescence is lost and pilus glycosylation is gained. (**B**) Time-lapse of de novo occurring wt clone within an expanding *G− green* colony. Full yellow line: front of the microcolony obtained from the brightfield image. Scale bar: 10 μm.

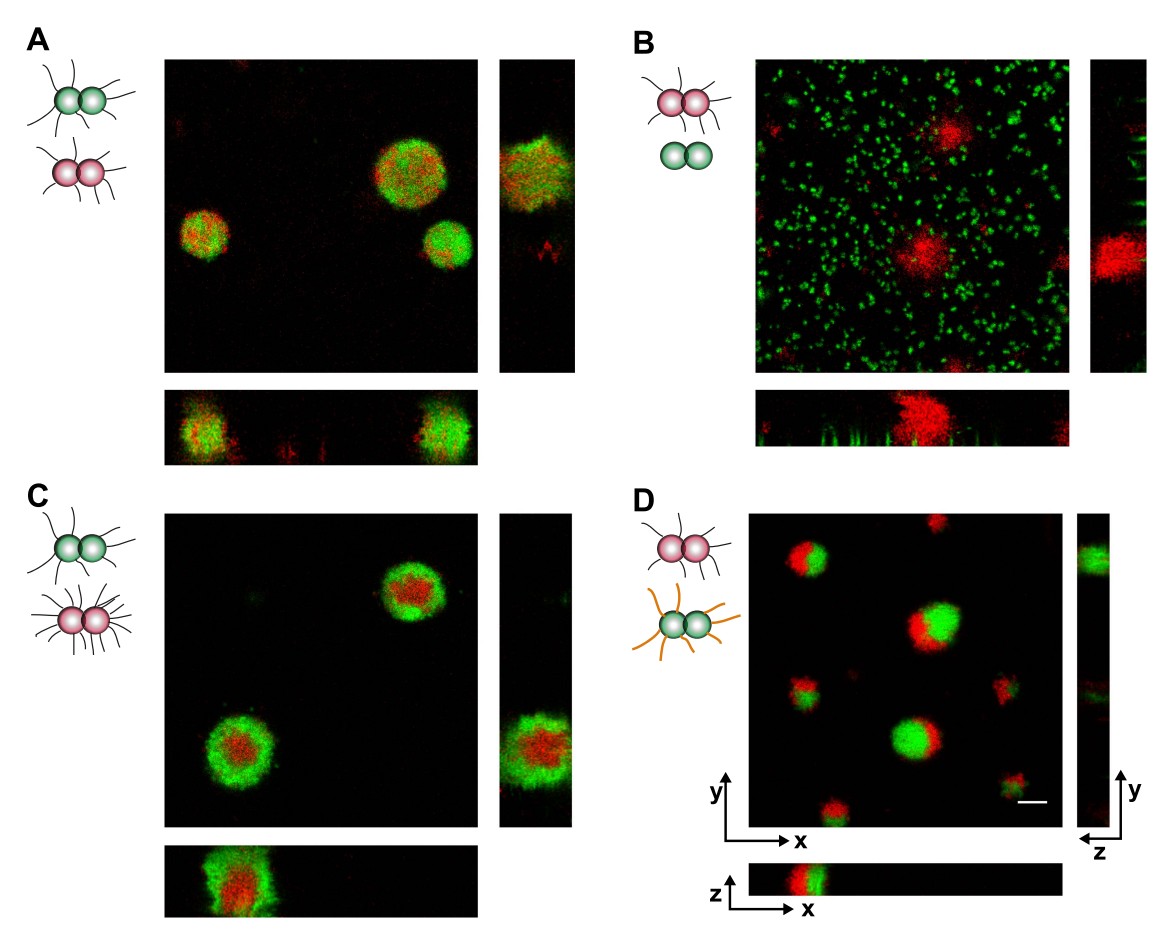

**Figure 7**. Loss of pili or gain of post-translational modification causes segregation in aqueous environment. Confocal stacks of gonococci mixed in a liquid and incubated on a glass surface for (3–5) hr. (**A**) *P+ red* was mixed with *P+ green*, (**B**) *P+ red* and *P− green*, (**C**) *P++ red* and *P+ green*, (**D**) *P+ red* and *G− green*. Scale bar: 10 μm.

bacteria did not mix, but only grew into separate clusters. Similarly, inhibition of glycosylation in one strain did not change the morphologies in a mix of *T−G− green* and *T− red* cells (*Figure 8B*) nor did they resemble the shape predicted by the DAH and the DSAH.

We conclude that active force generation by T4P is required for the cluster morphology predicted by the DAH and the DSAH.

## Discussion

In this study, we systematically investigated the effect of physical interactions between bacterial cells on sorting and cluster morphology. We demonstrate that type IV pili of the gonococcus are a versatile system for studying bacterial sorting.

### The DSAH predicts the morphologies of mixed microcolonies

The differential adhesion hypothesis, by Malcom Steinberg, predicts the morphologies of cell-clusters formed by two types of cells with different surface adhesins (*Steinberg, 1963*). The morphologies we observed in liquid culture are in accordance with the predicted ones. However, active pilus retraction was required for generating these well-defined morphologies. Recently, we provided evidence that gonococci move over surfaces by a tug-of-war of multiple pili (*Marathe et al., 2014*). Therefore, it is likely that bacteria within microcolonies employ a tug-of-war mechanism for segregation (*Figure 9*). Based on the pilus–pilus rupture forces, all four morphotyes shown in *Figure 7* can be explained. Bacteria with the same composition and number of pili exhibit equal net rupture forces between

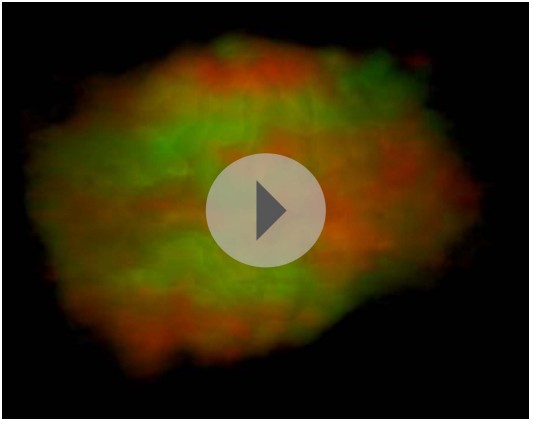

**Video 4.** Confocal reconstruction of P+ green and P+ red.

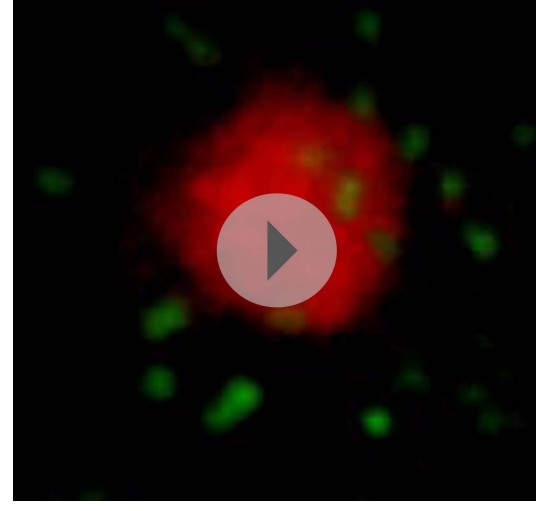

**Video 5.** Confocal reconstruction of P+ red and P−
green.

all cells. Accordingly, we found well-mixed aggregates. Bacteria without pili show no interaction amongst themselves or with wt bacteria, in agreement with $F_{wt/P-} \approx F_{P-/P-} \approx 0$. Co-incubation of hyperpiliated $P++$ and wt led to the formation of a sphere of $P++$ in the center and a fully surrounding shell of $P+$ in agreement with $F_{P++/P++} > F_{P++/wt} > F_{wt/wt}$. Finally, a mix of $P+$ and glycosylation-inhibited $G-$ bacteria with $F_{G-/G-} > F_{wt/wt} > F_{G-/wt}$ formed a sphere of $G-$ bacteria and with a crescent of $P+$ partially surrounding it. To conclude, the mixed morphologies coincide with the prediction of the DSAH (*Harris, 1976*).

Adding to the above, active forces have been shown to be important for cell sorting during embryonic development. In developing tissues, cortical contractile forces and cell adhesion act antagonistically, opposing each other with contraction tending to minimize and adhesion to maximize the contact area (*Fagotto, 2014*). In comparison, the cell body in our bacterial system can be considered as a point-object and the forces of pilus retraction are always attractive.

## Naturally occurring variation of pilus–pilus interactions are likely to induce sorting in gonococcal biofilms

Cell sorting as a consequence of variations in density and post-translational modification is very likely to affect the biology of pathogenic *Neisseria* species. *Neisserial* populations generate micro-heterogeneity via phase and antigenic variation of T4P-associated genes (*Rotman and Seifert, 2014*). The detection of multiple

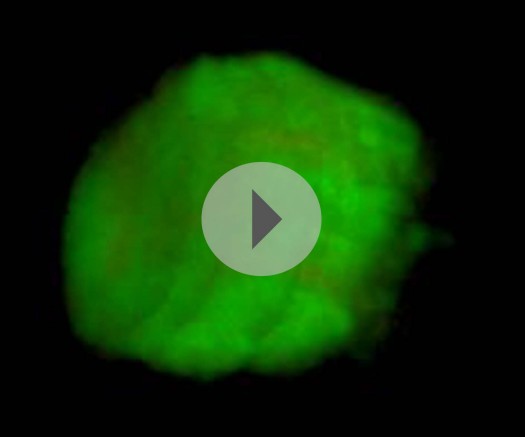

**Video 6.** Confocal reconstruction of P++ red and P+ green.

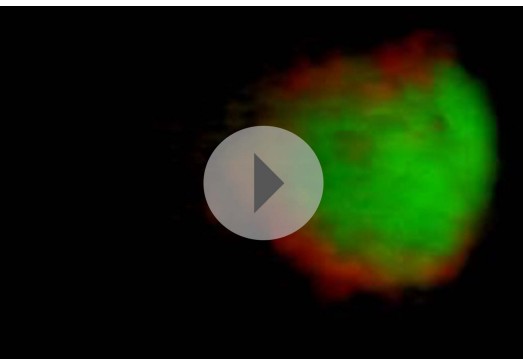

**Video 7.** Confocal reconstruction of P+ red and G− green.

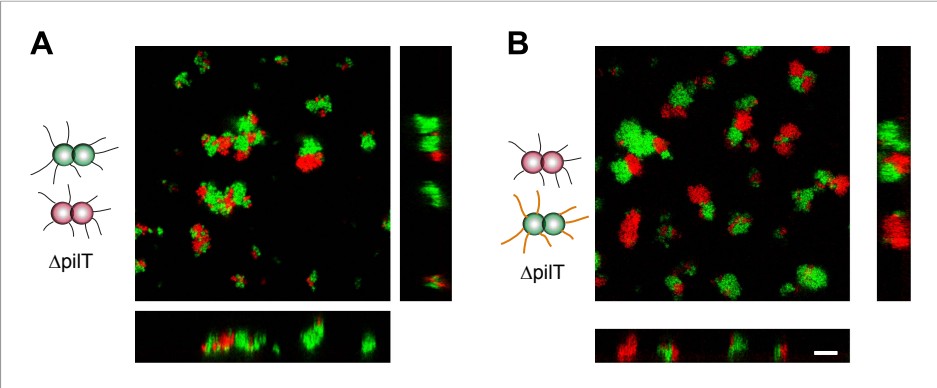

**Figure 8**. Active T4P retraction affects colony morphology. Confocal stacks of gonococci deficient in active force generation by T4P retraction were mixed in a liquid and incubated on a glass surface for (3–5) hr. (**A**) *T− red* was mixed with *T− green*, (**B**) *T− red*, and *T−G− green*. Scale bar: 10 μm.

antigenic variants in the sequence of the major pilin after 2 days of biofilm growth highlights this (*Kouzel et al., 2015*). Apart from the pilin gene, other genes affecting post-translational modification and pilus density are phase-variable (*Marri et al., 2010*). We propose that phase and antigenic variation trigger cell sorting and thereby affect the architecture of biofilms.

## Conclusion

We have tested the hypothesis that differential physical interactions between bacteria cause cell sorting in early biofilms. We generated a toolbox of gonococcal type IV pili with varying pilus density and pilus–pilus rupture forces. The morphotypes of mixed microcolonies were in remarkable agreement with the predictions of the DSAH proposing that cells sort on the basis of differential net rupture forces. Likewise to embryonic development, our findings suggest mechanical forces govern cell sorting in early biofilm formation.

## Materials and methods

### Bacterial growth conditions

*N. gonorrrhoeae* was grown overnight at 37°C and 5% $CO_2$ on agar plates containing gonococcal base agar (10 g/l Bacto agar [BD Biosciences, Bedford, MA, USA], 5 g/l NaCl [Roth, Darmstadt, Germany], 4 g/l $K_2HPO_4$ [Roth], 1 g/l $KH_2PO_4$ [Roth], 15 g/l Bacto Proteose Peptone No. 3 [BD], 0.5 g/l

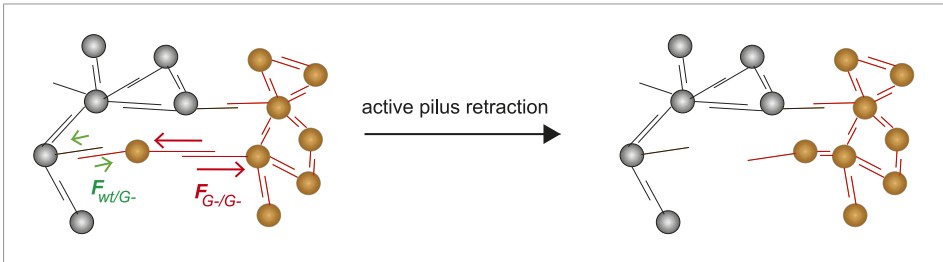

**Figure 9**. Model for tug-of-war mechanism of cell sorting. Pili form contacts between bacteria. For simplicity, only single pilus–pilus bonds between bacteria are shown. Pili continuously elongate and retract. During retraction, they generate force on an attached object. When pili bind to other pili and retract they generate force on each other. The probability of bond rupture increases with force. Since the rupture force between *G−* and *G−* pili is larger than between wt and *G−* pili, the latter bond is more likely to rupture. Pilus retraction pulls the cell body in the direction in which the strongest, least breakable bonds are formed. Cells whose bonds are most easily broken are squeezed outward to the peripheral layer.

soluble starch [Sigma–Aldrich, St. Louis, MO, USA]), and the following supplements: 1 g/l D-Glucose (Roth), 0.1 g/l L-glutamine (Roth), 0.289 g/l L-cysteine-HCL $\times$ H$_2$0 (Roth), 1 mg/l thiamine pyrophosphate (Sigma–Aldrich), 0.2 mg/l Fe(NO$_3$)$_3$ (Sigma–Aldrich), 0.03 mg/l thiamine HCl (Roth), 0.13 mg/l 4-aminobenzoic acid (Sigma–Aldrich), 2.5 mg/l β-nicotinamide adenine dinucleotide (Roth), and 0.1 mg/l vitamin B$_{12}$ (Sigma–Aldrich). Before each experiment, gonococcal colonies were resuspended in GC-medium.

## Strain constructions

All strains in this study were generated into the same VD300 background by transformation with either newly created plasmids or gDNA of existing strains.

The strains with the fluorescent reporter in the *igA1* locus were constructed as follows: the promoter region of *pilE* (P$_{pilE}$) with a SacI restriction site was amplified from gDNA of gonococcal strain MS11 using primers NK83 and NK135. *gfpmut3* with a SacI restriction site was amplified from the pAM239 using primers NK134 and NK133. A PCR fusion between P$_{pilE}$ and *gfpmut3* was generated using NK83 and NK133. The product was inserted into SacI site of p2/16/1 (*Wolfgang et al., 2000*), resulting in p*Iga::P$_{pilE}$ gfpmut3*. This plasmid was used to introduce the *gfpmut3 ermC* alleles into the *iga* locus of strain VD300, generating strain Ng105 *iga::P$_{pilE}$ gfpmut3 ermC*.

The mcherry fluorescent reporter strains were constructed using an integration vector for insertion of alleles between *lctP and aspC* on the gonococcal chromosome, following a previously reported approach (*Mehr and Seifert, 1997*). *aspC* was amplified from chromosomal DNA using NK63 and NK64 introducing an engineered MCS at 5′ end and a HindIII at 3′ end. *lctP* was amplified using NK61 and NK62 introducing a HindIII restriction site at the 5′ end and a MCS at 3′ end. The fragments were fused with primers NK61 and NK64 and inserted into *Hind*III site of pUP6, resulting in pLA. The aminoglycoside-3′ adenyltransferase (*aadA*) gene, encoding for a streptomycin/spectinomycin adenyltransferase, was amplified from R100.1 plasmid with primers NK 59 and NK 60 and cloned into a unique *Sac*I site in the MCS of pLA, resulting in pLAS. P$_{pilE}$ was amplified using NK23 and NK46 generating a FseI restriction site. *mcherry* was amplified from pSP64 with NK19 and NK51 generating a PacI restriction site. The PCR fusion between P$_{pilE}$ and *mcherry*, amplified with NK46 and NK51, was inserted between FseI and PacI sites of pLAS resulting in pLAS::P$_{pilE}$ mcherry.

This plasmid was used to introduce the *mcherry aadA* alleles between lctP and aspC of strain VD300, generating strain Ng106 *lctP:mcherry aadA:aspC*. Furthermore, this plasmid was used to introduce mcherry into strain N400 (*Freitag et al., 1995*), finally yielding strain Ng116 *recA6ind(tetM); lctP: P$_{pilE}$ mcherry aadA:aspC*. Finally, the plasmid was introduced in Ng109 *igA1::pilE pilE ermC*, which in turn was generated by transformation with gDNA of N400 *pilE+++* (*Holz et al., 2010*), yielding strain Ng110 *igA1::pilE pilE ermC; lctp: P$_{pilE}$ mcherry aadA:aspC*.

The Ng081 *pilE::PpilE gfpmut3 kan strain was constructed using* a PCR fragment that contains fluorescent gene reporter with a selectable drug marker flanked by ends of the target gene. Individual fragments were amplified using the primers listed in *Table 3* by PCR either from chromosomal DNA of gonococcal strain MS11 or from plasmid p*Iga::P$_{pilE}$ gfpmut3*. The 5′ flanking region of *pilE* and the region of gfp were amplified from DNA from plasmid p*Iga::P$_{pilE}$ gfpPmut3* with primers KH1a and KH4. The kanamycin selectable marker was amplified from plasmid *pUP6* using KH5 and KH6. The 3′ flanking region of *pilE* was amplified from chromosomal DNA using KH7 and KH8. The amplified fragments were fused yielding the final fragment *pilE::gfpmut3 kan*, which was used for transformation.

The Ng095 *pglF::PpilE gfpmut3 kan strain was constructed using* a PCR fragment that contains fluorescent gene reporter with a selectable drug marker flanked by ends of the target gene. Individual fragments were amplified using the primers listed in *Table 3* by PCR either from chromosomal DNA of gonococcal strain VD300 or from plasmid p*Iga::P$_{pilE}$ gfpmut3*. The 5′ 2flanking region of *pglF* was amplified from chromosomal DNA with primers KH34 and KH35. P$_{pilE}$ was generated using the PCR from chromosomal DNA with KH36 and N135. The *gfpmut3* reporter gene was amplified from plasmid p*Iga::P$_{pilE}$ gfpmut3* with N134 and KH4. The kanamycin selectable marker was amplified from plasmid p*Iga::P$_{pilE}$ gfpmut3* using KH5 and KH37. The 3′ flanking region of *pglF* including start of *pglB* was amplified from chromosomal DNA using KH38 and KH39. The amplified fragments were fused yielding the final fragment *pglF::gfpPmut3-kan*, which was used for transformation.

Table 3. Primers used in this study

| Primers | Sequence 5′–3′ |
| --- | --- |
| KH1a | ATGCCGTCTGAATTCCGACCCAATCAACACACC |
| KH4 | GTTCAATCATATGTGACCTCCTCTATTTGTATAGTTCATCC |
| KH5 | TAGAGGAGGTCACATATGATTGAACAAGATGGATTGC |
| KH6 | TCACTTACCGTCAGAAGAACTCGTCAAGAAGG |
| KH7 | TTCTTCTGACGGTAAGTGATTTCCCACGG |
| KH8 | ATGCCGTCTGAACGCACCGATATAGGGTTTG |
| KH34 | AAAAAGAATTCATGCCGTCTGAAGCAAAATCGACCTGCACCATCTGAT |
| KH35 | CGGGTGTGTTGATTGGGTCGGTTTTGATGTCCGGTCGGCGGC |
| KH36 | GCCGCCGACCGGACATCAAAACCGACCCAATCAACACACCCG |
| KH37 | GAACCGACATAGAAGTAGTCAGGATGATTTTCAGAAGAACTCGTCAAGAAGGCG |
| KH38 | CCTTCTTGACGAGTTCTTCTGAAAATCATCCTGACTACTTCTATGTCGGTTC |
| KH39 | AAAGGATCCATGCCGTCTGAACATCAAAAGCGGGCGGGGG |
| NK83 | TTGAGTCTTCCGACCCAATCAACACACCCGATAC |
| N135 | AGTTCTTCTCCTTTACGCATAAAATTACTCCTAATTGAAAGGG |
| N134 | TTTCAATTAGGAGTAATTTTATGCGTAAAGGAGAAGAACTTTTCAC |
| NK133 | TTGAGCTCCTATTTGTATAGTTCATCCATGCC |
| NK19 | TTAGGAGTAATTTTATGGTGAGCAAGGG |
| NK23 | TCGCCCTTGCTCACCATAAAATTACTC |
| NK46 | CATTGGCCGGCCTTCCGACCCAATCAACACACC |
| NK51 | CATTAATTAATTACTTGTACAGCTCGTCCATGCC |
| NK60 | TTCGGTCTCCACGCATCGTCAG |
| NK61 | TTTAAGCTTATGGCACTTTTCCTCAGCATATTCCC |
| NK62 | GAGCTCTTAATTAAATGCATGGCCGGCCCTAGAGGAAGAAAATCATTGCCGCGAC |
| NK63 | GGCCGGCCATGCATTTAATTAAGAGCTCATGTTCTTCAAGCACATCGAAGCC |
| NK64 | TTTAAGCTTTTACAAGACTTTCACGATGCTTTCGC |
| NK65 | TTTTAATTAAATGCGTAAAGGAGAAGAACT TTTCACTGG |
| NK66 | GTAAGGCCGGCCCTATTTGTATAGTTCATCCATGCCATGTGTAATC |

Strains Ng119, Ng120, and Ng121 were generated as follows. gDNA of GT17 *pilT::m-Tn3cm* was used to transform *P+ green*, *P+ red*, and *G− green*, respectively, yielding *T− green*, *T− red*, and *T−G− green*.

Strains Ng118 and Ng116 were generated as follows. The plasmid *pIga::P$_{pilE}$ gfpmut3* was used to transform N400 (*Freitag et al., 1995*), yielding *P+ red\**. The resulting N400 *igA1::P$_{pilE}$ gfpmut3 ermC* was then transformed with gDNA of Ng055 pilQ::m-Tn3cm, yielding *P$^Q$− green*.

Direct DNA sequencing of PCR products derived from gonococcal transformants was performed by GATC Biotech AG (Konstanz, Germany) to verify the insertions and the absence of any other alterations.

## Colony growth and single-cell transformation on agar

gDNA was isolated with the DNeasy Blood & Tissue kit (Qiagen, Hilden, Germany) from strains that already carry the genes for the fluorescent protein in the locus of interest. Individual colonies of non-fluorescent and fluorescent bacteria were picked from overnight plates and highly diluted in GC-medium yielding a gonococcal suspension with mainly dark and a few fluorescent bacteria. The latter served as a control to analyze the generation time via the rate of fluorescence increase. 4 µl gDNA solution and 1 µl gonococcal suspension were mixed and immediately used to inoculate a drop onto the center of a GC-plate. The drop was allowed to dry and the plate was mounted onto an inverted

epi-fluorescence microscope (Nikon Ti-E, Japan) equipped with a motorized stage and a custom-built thermo-box. The plate is held in a custom-built chamber with a glass bottom to allow for imaging and $CO_2$ supply, keeping the plate under an atmosphere of 37°C and 5.9% $CO_2$. Time-lapse imaging was done employing a 40×/0.6 long-working distance air objective (Nikon) capturing images every 15 min over a grid of 120 different positions yielding an imaging area of 2.3 by 2.5 mm.

## Range expansion

*P+ red* and *P$^Q$− green* colonies were grown overnight on GC-plates. A small suspension with mostly P+ red and a few P$^Q$− green was mixed in GC-medium. A spot of 5 µl of the suspension was used to inoculate GC-plates, yielding a fully close front of bacteria. The range expansion of growing P+ red competing with PQ− green was monitored using the time-lapse microscopy described for the single-cell transformation assay on agar. It should be noted that the higher number of cells in a range expansion inevitably leads to the presence of stochastically varied non-piliated cells. In order to decrease this effect and focus on piliated red cells and non-piliated green cells, strains in a $recA_{ind}$ background were employed, which highly reduces the chance of spontaneously occurring *P−* cells (*Criss et al., 2005*).

## Image analysis

Time-lapse epi-fluorescence microscopy under an atmosphere with enriched $CO_2$ at 37°C allowed to take videos of gonococcal micro-colonies growing from individual cells (*Figure 5—figure supplement 1A*). The transformation of gDNA could be visualized by detection of fluorescence resulting from the expression of fluorescent proteins (*Figure 5—figure supplement 1D*). The recorded videos were processed as follows:

Images were taken over a grid of 120 positions. A normalized, smoothed, average of the images of all positions at the beginning of the experiment is used to correct for inhomogeneous illumination. At the beginning only single cells are present, hence only the average illumination is visible in the average image. All positions of a single time-point were stitched together. High local intensity variance in windows of 5× 5 pixels served as indicator for the presence of a growing microcolony, that is, there is a high variation of dark and bright pixels for colonies, but only small variance for pixels without colonies (*Figure 5—figure supplement 1B*). Thresholding the local variance image gives a binary image indicating the growing colonies. The binary image was further processed by binary dilations, to ensure connected segmentations of colonies, which was followed by binary erosions to reduce any segments due to noise, but, also, to shrink the contour of the segment of a colony, such that the contour lies more on top of the bacteria at the front. After filling the holes in all binary segments, the contour pixels were defined to be the pixels that are affected by a further binary erosion. A circle fit on the contour pixels of each individual segment gives the information of center and radius of the colony. The centers of all circles were tracked to find regions of interests (ROIs) of growing microcolonies that have not yet grown into another colony. Fluorescence was segmented as follows: the stitched fluorescence image was convolved with a gaussian kernel of 11 × 11 px with σ = 1.5 px. The local intensity background of each ROI was found and subtracted. Thresholding of the background corrected fluorescence inside every ROI allowed to segment the patches of fluorescent bacteria (*Figure 5—figure supplement 1*).

The radii of the colony turned out to grow exponentially for at least 10 hr. Further image analysis was restricted to this time-period. Given the center and radius of the growing colony at any time, all Cartesian coordinates can be transformed into polar coordinates. The exponential growth of the radius of the colony implies that all distances inside the colony should, also, grow in the same exponential fashion. This can be illustrated by normalizing the polar coordinates by the radius. *Figure 5—figure supplement 1C,D* show a row of brightfield and fluorescence images, with the circle fit of the colony to fill always the same proportion of the sub-image. As a result, the scale bar shrinks in time, while the apparent colony size stays constant. Due to this normalization, the apparent position of segmented patches, also, stays roughly constant. The last column in (b) shows a superposition of all time-points of the normalized coordinates onto a unit-circle at the top, and a superposition of all time-points without normalization at the bottom onto the final fluorescence image. Due to the clear separation of clonal normalized coordinates, a manual selection of areas inside the unit-circle allowed to easily define all coordinates resulting from the same clone.

The spatial variance $\sigma^2$ can now be calculated using the information of background-corrected intensities and the Cartesian and polar coordinates for each individual clone for all time-points. To this end, the total variance $\sigma^2$, the orthogonal variances $\sigma_x^2$ and $\sigma_y^2$ in Cartesian coordinates are expressed as follows:

$$\sigma^2 = \sigma_x^2 + \sigma_y^2,$$

$$\sigma_x^2 = \sum_{i=1}^{N} x_i^2 \frac{I_i}{I_{tot}} - \left( \sum_{i=1}^{N} x_i \frac{I_i}{I_{tot}} \right)^2,$$

$$\sigma_y^2 = \sum_{i=1}^{N} y_i^2 \frac{I_i}{I_{tot}} - \left( \sum_{i=1}^{N} y_i \frac{I_i}{I_{tot}} \right)^2,$$

$$I_{tot} = \sum_{i=1}^{N} I_i.$$

Our experiment runs during the exponential growth phase. Following a transformation event, the number of fluorescent proteins per cell generated by the offspring of a single transformant is expected to increase and eventually saturate at an equilibrium concentration. Thus, before saturation, the overall fluorescence initially increases more quickly, but then tends to increase exponentially (*Figure 5—figure supplement 1E*). In our experiments, a fraction of the growing colonies originated from initially fluorescent cells. These colonies expressed the fluorescent reporter at its equilibrium concentration right from the start of the experiment and the rate of fluorescence increase was used to measure the generation time.

The development of the total fluorescence $I_{tot}$ of each clone is used to deduce the number of bacteria within a clonal patch. To this end, the rate of fluorescence increase $I_{tot}(t)$ can be modeled assuming a constant rate increasing the fluorescence per cell, $k_{prod}$, and degradation of fluorescence, $k_{deg}$, with N being the number of fluorescent cells:

$$\frac{dI_{tot}}{dt} = k_{prod} N - k_{deg} I_{tot}.$$

Thus, the total fluorescence with $I_0$ being the equilibrium intensity of a single cell can be fitted as follows:

$$I_{tot}(t) = I_0 N (t - t_0) \left( 1 - e^{-2 k_{deg}(t - t_0)} \right),$$

where: $N(t) = 2^{t/\tau_{growth}}$

$I_0$, $t_0$, and $k_{deg}$ are fit parameters. The value for $\tau_{growth}$ is taken from the rate of fluorescence increase of those growing colonies, which were fluorescent right from the beginning of the experiment. These colonies expressed the fluorescent reporter at or close to its equilibrium concentration and the rate of fluorescence increase was used to measure the generation time, $\tau_{growth}$. The model $I_{tot}(t)$ fits well onto the data (*Figure 5—figure supplement 1E*), such that the time at which the clone arose, $t_0$, together with the generation time, $\tau_{growth}$, allows to estimate the number of fluorescent bacteria $N(t - t_0)$.

## Confocal microscopy

Gonococcal microcolonies were cultivated at 37°C in *ibidi* μ-Slides I$^{0.8}$ Luer. For microcolony development in a continuous-flow chamber, GC-medium with 1% IsoVitaleX was used. Bacteria from overnight plates of each strain were re-suspended in GC to an optical density at 600 nm (OD$_{600}$) of 0.1 and two hundred microliters of each culture were inoculated into flow chambers and left for 1 hr at 37°C to allow attachment to the glass surface. After 1 hr, the flow was resumed and pumped through the flow cells at a flow rate of 3 ml/hr for each channel (0.2 mm/s) by using a peristaltic pump (model 205U; Watson Marlow, Calmouth, UK).

Gonococcal microcolonies in aqueous environment were visualized by using a Confocal Laser Scanning Microscope (Nikon Ti-E C1). Images were obtained using a 60×/1.2 water objective (Nikon) (*pinhole* size of 60 μm). Dual fluorescence emission was observed using an argon-ion laser with 488 nm for green fluorescent protein and a 543-nm filter of HeNe laser for mCherry fluorescence. To

avoid cross-talk between colors, the images were acquired sequentially, each only with corresponding laser excitation. From each fluidic channel, five to six image stacks were acquired randomly down through the channel. 3D visualization of image was done using ImageJ.

## Measurement of pilus–pilus rupture force

The optical tweezers were assembled on a Zeiss Axiovert 200 microscope as described previously (*Clausen et al., 2009*; *Anderson et al., 2014*). The assay was modified for measuring the force generated by a single surface-attached bacterium as a function of time (*Anderson et al., 2014*; *Dewenter et al., 2015*). The trap was calibrated by power spectrum analysis of the Brownian motion of individual monococci assuming a bacterial diameter of 1 µm and was found to have a stiffness of 0.14 pN/nm ± 10%. The position of trapped bacteria was detected at 20 kHz using a quadrant photodiode and the sample stage was movable via a combined piezo and electric motor.

We selected for monococci, that is, round bacteria since force measurement could be performed with higher accuracy than with diplococci. Using the drag force method, we found that the linear force range of the optical trap was $d$ = 450 nm when gonococci were trapped. Therefore, the maximum force we detected was $F$ = 65 pN. The retraction experiments were performed in a sealed chamber using a low-density suspension of gonococci, allowing for undisturbed measurement of single bacteria over a period of minutes. During the experiment, a single bacterium was trapped near the surface using the optical tweezers. Retraction of surface-bound pili resulted in deflection of the bacterium from the center of the optical trap as measured by the four-quadrant photodiode. The force acting on a bacterium is proportional to its deflection and was calculated using the calibration described above. After several minutes, a detrimental effect of the optical trap on the bacterium became apparent and it was abandoned.

## Acknowledgements

We gratefully thank Michael Koomey and Hank Seifert for the donation of bacterial strains and vectors, and Oskar Hallatschek and the Maier group for helpful discussions. This work was supported by the Deutsche Forschungsgemeinschaft through grants SFB 680 and MA3898.

## Additional information

### Funding

| Funder | Grant reference | Author |
| --- | --- | --- |
| Deutsche Forschungsgemeinschaft (DFG) | SFB680 | Nadzeya Kouzel, Katja Henseler, Berenike Maier |
| Deutsche Forschungsgemeinschaft (DFG) | MA3898 | Nadzeya Kouzel, Lena Dewenter |

The funders had no role in study design, data collection and interpretation, or the decision to submit the work for publication.

### Author contributions

ERO, NK, Conception and design, Acquisition of data, Analysis and interpretation of data, Drafting or revising the article; LD, Conception and design, Acquisition of data, Analysis and interpretation of data; KH, Conception and design, Acquisition of data; BM, Conception and design, Analysis and interpretation of data, Drafting or revising the article

### Author ORCIDs

Enno R Oldewurtel, http://orcid.org/0000-0002-2813-0259

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
