## [Decision Letter]

Thank you for submitting your work entitled “Differential interaction forces govern bacterial sorting in early biosfilms” for peer review at *eLife*. Your submission has been favorably evaluated by Naama Barkai (Senior Editor), and two reviewers, one of whom, Roberto Kolter is a member of our Board of Reviewing Editors.

The reviewers have discussed the reviews with one another and the Reviewing editor has drafted this decision to help you prepare a revised submission.

This is a resubmitted manuscript that was previously rejected. In our rejection letter we acknowledged that there was a very interesting story that had been poorly presented. In the revised version the authors have extensively revised their presentation. Indeed, they followed our recommendations very well. For one, they now start with presentation of the theory and proceed to use the *Neisseria* pili system to test it. They also have removed a lot of the results they had previously presented on phase variation because they detracted and confused more that they contributed to the main aim. Thus, we feel that the new version is much improved. The data that are now included are otherwise the same as before and they remain strong. Our main comments are that the paper needs editing. In particular, parts of the Discussion are redundant with the Results section (the parts that merely summarize the results again without additional discussion, e.g. the second and third paragraphs of the subsection “Differential physical interactions drive cell sorting”). These parts should be removed. Also, we think there was an error in the equations in the subsection “Image analysis” (one of the equation still seems incorrect; please see attachment).

[Editors’ note: a previous version of this study was rejected after peer review, but the authors submitted for reconsideration. The previous decision letter after peer review is shown below.]

Thank you for choosing to send your work entitled “Genetic variation alters pilus–pilus interaction force and governs gonococcal cell sorting” for consideration at *eLife*. Your full submission has been evaluated by Naama Barkai (Senior Editor), and two reviewers, one of whom, Roberto Kolter, is a member of our Board of Reviewing Editors, and the decision was reached after discussions involving all three. Based on our discussions we regret to inform you that your work in its current format will not be considered further for publication in *eLife*.

However, the reviewers all agreed that the main problem with the manuscript was in its presentation and that, indeed, the results were of significance but that their importance was hidden. The work addresses a fundamental question (though this was not obvious until the last part of the manuscript), it uses an innovative approach to address this question and it provides conclusive results and answers. Thus, we would be willing to reconsider a completely re-written manuscript, where the hypothesis is presented first and then its test. You should note that if you decide to re-submit, the manuscript would be treated as a brand new submission and would undergo the full process of review.

All three reviewers found the structure of the manuscript very confusing. It was not until the end that it became evident that the work tested theoretical predictions of how differential cell–cell interaction forces drive spatial arrangement of progeny cells. By making this key link to the theoretical predictions only in the Discussion, the significance of the work is almost completely lost. In addition, the emphasis on the pathogen and antigenic and phase variation are such that the importance of the actual work is diminished. In our opinion, the solution is a complete overhaul of the manuscript to make it hypothesis driven. First present the theoretical predictions and then build on that to present the results of how cell–cell interaction forces influence segregation. The model system is a great tool but should not be the focus of the paper.

Reviewer #1 (Roberto Kolter):

The authors present a nice set of experiments that allow them to conclude that cell–cell interaction forces (which vary as a consequence of presence/absence/modification of type IV pili in the gonococcus) leads to cells “segregating” or “sorting”. From my perspective the experiments have been well carried out. A very important control was carried out in using cells that were mutated for the ability to vary their pili and sectoring was shown to be reduced. This is critical to me because the observed reduction (rather than complete lack) of segregation already indicates that a major contributor to segregation is indeed simply where the cells are and thus defines where their progeny will be located. So the while the forces of the cell–cell interactions are contributing to the segregation patterns observed, they are not the sole factor. This greatly dampens the claims regarding the biology of this pathogen in particular. But most importantly, from my perspective, while the finding that pilus–pilus interaction forces influence segregation is very “neat”, it is really not a great insight regarding the biology of the organism. The contribution, as currently presented, seems to be more appropriate for a more specialized journal. However, if the results were to be presented differently, first posing the importance of how theory predicts that interaction forces between cells (in general) should lead to cell progeny segregation and only after that proceed to present the findings, this could be an important contribution. This work is not so much about this pathogen in particular but in general for all biological situations where cell–cell interactions might influence the spatial arrangements of progeny cells. This significance in almost completely lost in the current format.

Reviewer #2:

1) In general, I found the text hard to read. This was a consequence of two aspects. First, I often did not get a good understanding of where the text was heading – I understood the content but did not understand the direction of the writing (for example in the third and fourth paragraphs of the Introduction). The second and related point is that the authors sometimes provided pieces of information at a moment where I did not yet know why this piece of information is relevant. One example: In the first paragraph of the subsection “Spatio-temporal dynamics of a variant with reduced pilus density within an expanding colony” the authors start discussing transformation, but it only becomes clear later that transformation plays a role in this study. I realize that my points concern writing style, which is a matter of personal preference – but I wanted to mention this in case it is useful for preparing a revised version. In short, a major re-write would be necessary.

2) The Introduction and the Results are very difficult to read. For example, the Results say that the experimentally determined interaction forces *explain* the observed spatial arrangement – but this is only true if one has the theoretical predictions in mind. Without knowing about these predictions at this point, it is not possible to see how exactly the interaction forces can explain the spatial arrangement. The authors do mention a link between interaction forces and spatial arrangement at the end of the Introduction and in the Results section (subsection “Interaction force between pilus-variants”). However, these statements are vague and understood only much later that these explanations are on solid theoretical ground.

3) Related to the point above, the main focus of the manuscript did not become immediately clear to me. Initially, the focus seemed to be on phase variation and type IV pili, but later the main focus shifted on fundamental questions about how differential cell–cell adhesion drives spatial arrangement of cells in a multi-cell context. I would find it easier if the study would just have one main focus.

4) Also related to the two previous points: because the link to the theoretical predictions (and therefore to the fundamental question) only comes so late in the manuscript, it did not become clear to me how much is already known about the link between interaction forces and spatial arrangement. Are there many other studies that address this issue? What are the main knowledge gaps? I would welcome more presentation of this early on in the manuscript.

5) As the authors show, the positioning of the mutants with fewer pili at the front of the expanding colony is probably a consequence of both their increased growth rate and of altered cell–cell interacting forces. Given that the manuscript focuses on the impact of interacting forces on spatial structure, it would seem important to try to estimate the relative importance of these two effects.

---

## [Author Response]

[…] In particular, parts of the Discussion are redundant with the Results section (the parts that merely summarize the results again without additional discussion, e.g. the second and third paragraphs of the subsection “Differential physical interactions drive cell sorting”). These parts should be removed.

We have shortened the Discussion and removed the redundant part.

Also, we think there was an error in the equations in the subsection “Image analysis” (one of the equation still seems incorrect; please see attachment).

We thank the reviewers for pointing this out. σ_φ_^2^ was determined as indicated in the Methods part. This quantity was intended to be used for characterization of the bacterial spread along the contour of the colony, once the *P*− bacteria was located at the front. We are aware that σ_φ_^2^ does not represent the variance of the polar coordinate φ. We realize, however, that we did not formulate this properly. This part of the analysis is not important for the conclusion of the work. Therefore, we decided to remove Figure 5—figure supplement 1 and its description from the manuscript.

[Editors’ note: the author responses to the previous round of peer review follow.]

[…] All three reviewers found the structure of the manuscript very confusing. It was not until the end that it became evident that the work tested theoretical predictions of how differential cell–cell interaction forces drive spatial arrangement of progeny cells. By making this key link to the theoretical predictions only in the Discussion, the significance of the work is almost completely lost. In addition, the emphasis on the pathogen and antigenic and phase variation are such that the importance of the actual work is diminished. In our opinion, the solution is a complete overhaul of the manuscript to make it hypothesis driven. First present the theoretical predictions and then build on that to present the results of how cell–cell interaction forces influence segregation. The model system is a great tool but should not be the focus of the paper.

Reviewer #1 (Roberto Kolter):

[…] But most importantly, from my perspective, while the finding that pilus–pilus interaction forces influence segregation is very “neat”, it is really not a great insight regarding the biology of the organism. The contribution, as currently presented, seems to be more appropriate for a more specialized journal. However, if the results were to be presented differently, first posing the importance of how theory predicts that interaction forces between cells (in general) should lead to cell progeny segregation and only after that proceed to present the findings, this could be an important contribution. This work is not so much about this pathogen in particular but in general for all biological situations where cell–cell interactions might influence the spatial arrangements of progeny cells. This significance in almost completely lost in the current format.

We thank Drs. Barkai and Kolter, and Reviewer #2 for their extremely helpful comments. Following their suggestions, we have re-structured the manuscript. The major changes are as follows:

1) All results about phase- and antigenic variation were removed. We feel that these experiments caused confusion at various levels.

2) The manuscript was re-structured to make it hypothesis-driven. The story now evolves around the question whether differential cell–cell interaction forces trigger cell sorting in early biofilms.

3) The question whether active force generation was important came up several times when talking to colleagues. We performed additional experiments showing that pilus retraction was necessary to form defined colony morphotypes. This finding strongly favors a tug- of-war as the underlying mechanism of cell sorting in agreement with the differential strength of adhesion hypothesis.

Reviewer #2:

1) In general, I found the text hard to read. This was a consequence of two aspects. First, I often did not get a good understanding of where the text was heading – I understood the content but did not understand the direction of the writing (for example in the third and fourth paragraphs of the Introduction). The second and related point is that the authors sometimes provided pieces of information at a moment where I did not yet know why this piece of information is relevant. One example: In the first paragraph of the subsection “Spatio-temporal dynamics of a variant with reduced pilus density within an expanding colony” the authors start discussing transformation, but it only becomes clear later that transformation plays a role in this study. I realize that my points concern writing style, which is a matter of personal preference – but I wanted to mention this in case it is useful for preparing a revised version. In short, a major re-write would be necessary.

The Introduction was re-written entirely. We have taken care that transformation is properly explained where it initially comes into play in the Results part.

*2) The Introduction and the Results are very difficult to read. For example, the Results say that the experimentally determined interaction forces* explain *the observed spatial arrangement – but this is only true if one has the theoretical predictions in mind. Without knowing about these predictions at this point, it is not possible to see how exactly the interaction forces can explain the spatial arrangement. The authors do mention a link between interaction forces and spatial arrangement at the end of the Introduction and in the Results section (subsection “Interaction force between pilus-variants”). However, these statements are vague and understood only much later that these explanations are on solid theoretical ground.*

We agree. The story was re-structured to make it hypothesis-driven.

3) Related to the point above, the main focus of the manuscript did not become immediately clear to me. Initially, the focus seemed to be on phase variation and type IV pili, but later the main focus shifted on fundamental questions about how differential cell–cell adhesion drives spatial arrangement of cells in a multi-cell context. I would find it easier if the study would just have one main focus.

The results concerning phase- and antigenic variation were removed and the manuscript was re-structured (see above).

4) Also related to the two previous points: because the link to the theoretical predictions (and therefore to the fundamental question) only comes so late in the manuscript, it did not become clear to me how much is already known about the link between interaction forces and spatial arrangement. Are there many other studies that address this issue? What are the main knowledge gaps? I would welcome more presentation of this early on in the manuscript.

The theoretical predictions are now described in the Introduction. All of the related studies that we are aware of are now described in the same section.

5) As the authors show, the positioning of the mutants with fewer pili at the front of the expanding colony is probably a consequence of both their increased growth rate and of altered cell–cell interacting forces. Given that the manuscript focuses on the impact of interacting forces on spatial structure, it would seem important to try to estimate the relative importance of these two effects.

We have performed various controls to show that differential pilus–pilus interaction forces are sufficient for cell sorting:

A) The pilin glycosylation inhibited strain has the same generation time as the wt but differential rupture forces. We found that glycosylated bacteria segregate from glycosylation-inhibited cells.

B) For the agar plate assay, we analyzed the mobility of non-piliated cells and compared it to piliated cells. To disentangle the effects of different generation times and different physical interactions, we plotted the location variance as a function of cell number rather than time. The fact that the variance was higher for non-piliated cells than for piliated cells shows that they are more mobile and have the potential to segregate.

C) We performed an additional experiment to verify that *P*− cells segregate from *P+* cell on agar plates based on differential physical interactions. We generated the strain *P*^*Q*^− that did not generate pili because it can’t generate the export channel for the pili. We show that the generation time of this mutant is comparable to the wt. In Figure 5—figure supplement 2 we show that *P*^*Q*^− segregates from *P+*.

D) In liquid environment microcolonies form, fuse, and reshape within minutes. This point was published previously by Dewenter et al. (Integr. Biol. 2015). This time-scale is much shorter than the generation time. Therefore we can expect a quasi-equilibrium structure of the mixed microcolony. In agreement with this assumption, we found nearly complete cell sorting.

To clarify the significance of these controls, we dedicated the section *“*Differential physical interactions drive cell sorting*”* of the Discussion to this topic.